# The Human Gut Virome and Its Relationship with Nontransmissible Chronic Diseases

**DOI:** 10.3390/nu15040977

**Published:** 2023-02-15

**Authors:** Shahrzad Ezzatpour, Alicia del Carmen Mondragon Portocarrero, Alejandra Cardelle-Cobas, Alexandre Lamas, Aroa López-Santamarina, José Manuel Miranda, Hector C. Aguilar

**Affiliations:** 1Department of Microbiology and Immunology, College of Veterinary Medicine, Cornell University, Ithaca, NY 14853, USA; 2Laboratorio de Higiene, Inspección y Control de Alimentos (LHICA), Departamento de Química Analítica, Nutrición y Bromatología, Universidade de Santiago de Compostela, 27002 Lugo, Spain

**Keywords:** Microviridae, Caudovirales, fecal viral transference, virome, obesity, diabetes, inflammatory bowel disease

## Abstract

The human gastrointestinal tract contains large communities of microorganisms that are in constant interaction with the host, playing an essential role in the regulation of several metabolic processes. Among the gut microbial communities, the gut bacteriome has been most widely studied in recent decades. However, in recent years, there has been increasing interest in studying the influences that other microbial groups can exert on the host. Among them, the gut virome is attracting great interest because viruses can interact with the host immune system and metabolic functions; this is also the case for phages, which interact with the bacterial microbiota. The antecedents of virome-rectification-based therapies among various diseases were also investigated. In the near future, stool metagenomic investigation should include the identification of bacteria and phages, as well as their correlation networks, to better understand gut microbiota activity in metabolic disease progression.

## 1. Introduction

The human gut microbiota (GM) is tightly connected to human health through a complex biological system that consists of bacteria, fungi, archaea, and viruses, among other components [1]. The bacterial composition of the GM has been widely investigated in the last two decades in terms of composition, diversity, functionality, and the relationship with human metabolic diseases [2,3]. However, much less attention has been given to other components of the GM, such as viruses. Recent works found that the number of virus-like particles (VLPs) in the gut ranges in a ratio of viruses to bacteria from approximately 0.1 to 10, suggesting that the number of viruses in the human body is close to the number of bacteria [4].

After the colonization of the intestine during birth, the human gut virome (GV) evolves steadily during infancy/childhood, entering a period of major changes during the first 2 years of life, and then stabilizes in later childhood [5,6]. In the first months of life, the GV is dominated by viruses that infect bacteria (phages), whereas eukaryotic viruses increase in both abundance and diversity throughout childhood [6]. In a healthy adult, the human GV comprises phages, viruses that infect other cellular microorganisms, viruses that infect human cells, and viruses derived from food and found in the digestive tract in a transient manner [4,6]. Among eukaryotic viruses, eukaryotic DNA viruses, such as those of the family *Circovidae*; RNA viruses, such as *Enteroviridae*, *Parechoviridae*, and *Picornaviridae*; and plant viruses, primarily of the family *Vigaviridae*, are frequently found in infants, as are some types of potentially pathogenic viruses, such as those belonging to the genera rotavirus, norovirus, and sapovirus [6].

With respect to phages, at the beginning of infancy, phages of the order *Caudovirales* (tailed phages) predominate, primarily those belonging to the families *Siphoviridae*, *Podoviridae*, and *Myoriviridae*. Subsequently, however, the presence of phages of the order *Caudovirales* decreases as the presence of phages of the family *Microviridae*, order *Petivirales* (icosahedral phages without tails), increases during the first 2 years of life [4,7,8]. Within the order *Caudovirales*, CrAss-type phages, a family of DNA-tailed phages that can infect bacteria of the phylum Bacteroidetes, are currently considered to be the most abundant phages in the adult human GV [5,9,10]. Such CrAss phages are rarely detected in the gastrointestinal tract of infants during the first month of life but subsequently increase in prevalence, accounting for more than 90% of the GV content in an adult human [6,11].

Progressive maturation of the infant GM leads to a reduction in GV abundance and diversity simultaneously compared with those observed for the bacteriome [1]. Subsequently, GV tends to resemble what it will be during the adult stage. Paralleling stability in the cellular microbiome [4], adult GV is usually stable over time, as evident from recent studies showing that >90% of recognizable viral contigs persisted in individuals over one year [1,4].

Most of the viruses included in the human GV are DNA viruses, whereas eukaryotic RNA viruses are rare in healthy individuals, and most of those that meet these characteristics are primarily plant-infecting viruses [4]. However, this lower presence of RNA viruses may not be as real as the background suggests. Thus, it should be noted that, in general, the RNA virome in the human gut is less studied than the DNA virome, as RNA viruses are substantially less stable in samples than DNA viruses, making their identification by metagenomic sequencing difficult [5]. Therefore, it is quite possible that the actual presence of RNA viruses is higher than that published for several environmental niches, including animal feces [5], and that their presence has been systematically underestimated due to technical reasons [4].

The fact that the majority of adult human GV components are phages is an inference because, in most cases, most of the sequences discovered in GV metagenomic sequencing assays do not align with any information present in existing datasets [5]. Therefore, the proportion of GV viruses that can be identified in a completely unambiguous way is extremely low and does not reliably represent all GVs. As the most abundant phage order, *Caudoviridales* includes phages that can infect all major bacterial phyla found in the gut: Firmicutes, Bacteroidetes, Proteobacteria, and Actinobacteria [2].

The human GV, which included both phages and eukaryotic viruses, can be modified in their richness and diversity by factors that depend on the subject’s lifestyle [5]. In this regard, different factors, such as geography, diet, genetics, or drugs ingested, can shape the human GV [4] (Table 1). A recent study [12] concluded that geography is the most important contributor to variations in human GV. In addition, other works have shown that diet type can modify GV in adults [4], as well as age [13]. The objectives of this manuscript were therefore to provide and update the state-of-the-art knowledge of the relationship between the GV and nontransmissible chronic diseases. To achieve this goal, a narrative literature search was conducted up to 10 July 2021 for the Web of Science and Scopus databases. The term “human gut virome” was searched in the field “title, abstract and keywords” in the case of Scopus and “topic” in the case of Web of Science. A total of 269 articles were found. The selection of articles will be limited to studies published in English, with no restrictions on the year of publication, although the most prominent articles are those published after 2018. The authors reviewed the titles and the abstracts. If the abstracts reported useful information, full texts were read, and if the pre-established eligibility criteria were met, they were included in the review. After selecting the articles that fell into the selected scope, a total of 87 articles were selected and included in the review.

## 2. Host-Gut Virome Interactions/Relationships

In general, phages have a narrow activity spectrum, with each phage affecting a small number of closely related bacterial species and, sometimes, only specific serotypes or specific strains within the same species [4]. This fact is important because GV plays a key role in modulating the bacterial populations that are part of the GM. However, since most intestinal phages remain poorly understood, it is often difficult to relate phages to the bacteria that are part of the host GM [17]. Phages exhibit four types of life cycles (including lytic, temperate/lysogenic, pseudolysogenic, and bacterial budding), with lytic and lysogenic life cycles being the two classical forms [5,18] (Figure 1). Another phage cycle is pseudolysogeny, also called the stationary phase of the phage in the host cell [19]. In this phase, there is neither multiplication of the phage genome as in the lytic cycle nor replication synchronized with the cell cycle of the host cell as in the lysogenic cycle [19]. This process usually takes place when the host cell encounters unfavorable conditions such as starvation and ends when the phage enters an actual lysogenic cycle or enters a lytic cycle when bacterial growth conditions improve [20]. This cycle seems to play an important role in phage survival, as bacteria in the natural environment often exhibit very slow growth or starvation [21]. Another cycle of phages is bacterial budding. This cycle is interesting, as phages are released through the bacterial cell membrane without causing lysis of the bacterium by a budding-like process, producing a chronic release of phages [22].

A phage can multiply through the lytic cycle when it kills the bacterium to release progeny phages or as a bacteria containing dormant phages (prophage) when it integrates its DNA into the bacterial chromosome (lysogenic cycle) [4,20]. In a healthy state, the human GV is primarily composed of temperate phages, and its replication switches from temperate to lytic during host inflammation or stress [18]. Because of the predominance of phages over eukaryotic viruses in the GV, as well as their role in regulating bacteriome composition and function, the phageome has been the focus of most human GV research [20].

In addition to phages, the healthy human intestine often contains low proportions of eukaryotic viruses, which include occasionally detected DNA viral lineages such as *Anelloviridae*, *Geminiviridae*, *Herpesviridae*, *Nanoviridae*, *Papillomaviridae*, *Parvoviridae*, *Polyomaviridae*, *Adenoviridae*, and *Circoviridae* [21]. Eukaryotic RNA viruses primarily comprise multiple plant-related viruses [12], while the most abundant eukaryotic RNA virus infecting animals in the GV belongs to the *Picornaviridae* family [22].

GV plays a highly relevant role in autoimmune and inflammatory intestinal diseases [23]. Therefore, colonization of the intestine by eukaryotic viruses is very important for the maintenance of proper intestinal homeostasis and the development of host immunity. Eukaryotic viruses can increase intestinal inflammation in mice by sensing viral RNA by host Toll-like receptors 3 and 7 and downregulating interferon beta secretion [20], which protects the host from inflammation [23]. Widespread phage predation, lysogeny, and gene transfer also exert an important role in controlling density, diversity, and network interactions within gut-associated symbiotic bacterial communities [8,16]. In addition, phages interact with host innate immunity and cytokine synthesis, and importantly, short-chain fatty acids promote phage production in bacteria [24]. Another important function is providing a direct defense against bacterial invasion in the mucin layer, as well as the interaction with the human immune system to maintain immune homeostasis and reduce the disease process [5]. In addition, several factors encoded by phages may also affect the pathogenicity of intestinal bacteria by promoting their adhesion, invasion, colonization, and toxin production and delivery [5].

## 3. Gut Virome Determination

Currently, the composition of VLPs is primarily investigated through metagenomics, where high-throughput sequencing is computationally processed to construct genomes of de novo uncultured viruses [5]. The gastrointestinal tract is a habitat with enormous viral colonization, reaching 109 VLPs per g of intestinal content [1].

The assembly of viral genomes is a highly difficult computational task and produces fragmented assemblies and chimeric contigs [1]. However, the identification of viruses without enrichment from bulk metagenomics is increasingly used and overcomes the biases of size filtration steps while allowing the identification of primarily templated but also lytic viruses [1].

Studies on human gut phages are currently limited to relatively low taxonomic levels. Although several human GV databases have been developed in the last 2 years, most bacterial gut viruses (81–93%) are new and cannot be assigned a taxonomic position [8]. Therefore, although these databases have greatly expanded our knowledge of the quality and diversity of human gut viral genomes and provided informative annotation datasets [25], there are still significant barriers that do not allow a complete understanding of human GV composition and dynamics [26]. Unlike bacteria, where 16S rRNA gene sequencing represents a unique tool to identify any bacterial species using a single criterion, DNA viruses have no component that allows similar identification. RNA viruses contain the conserved gene RNA-dependent RNA polymerase (RdRP), which allows broad viral identification [27]. However, the lack of a virome database with comprehensive annotations is a major concern in achieving broad identification of the RNA virome.

Therefore, one of the most critical shortcomings of the metagenomic approach to studying the GV is the large discrepancy between the demonstrable diversity of gut viruses and the number of known human gut-associated bacteriophage genomes, as >80% of viral sequences do not match closed reference databases [8].

## 4. Relationship between the Gut Virome and Metabolic Pathologies

Although studies of the relationship between the human GM and metabolic diseases have primarily focused on bacteria, in recent years, the relationship between GV and some metabolic diseases has started to be investigated, with increasing emphasis on phages. In this regard, variations in populations of some specific viral families and genera have been shown to be related to the development and progression of some metabolic diseases [28,29,30,31]. For example, the high presence of the *Mimiviridae* family in the human gut might be associated with obesity and diabetes [31]. Human adenovirus infection was identified as a significant risk factor for the progression of nonalcoholic fatty liver disease (NAFLD) [32]. Furthermore, in liver cirrhosis, GV alterations correlate with cirrhosis progression [33].

The most widely investigated matter is the relationship between the GM and intestinal diseases, primarily inflammatory bowel disease (IBD) [1,34,35,36,37,38,39,40,41,42], although there is also a potential relation between GV and type 1 diabetes (T1D) [28,43,44], type 2 diabetes (T2D) [45], obesity [31,46,47], hypertension [48], malnutrition and low growth rate [4,37], metabolic syndrome [49], liver diseases [33,50,51], colorectal cancer (CRC) [52,53,54,55], melanoma [56], cognitive maintenance [46], and cerebral ischemia [24] (Table 2).

### 4.1. Metabolic Syndrome

A variety of conditions that occur simultaneously and increase the risk of heart disease, stroke, and T2D are referred to as metabolic syndrome. These conditions include increased blood pressure, hyperglycemia, excess body fat around the waist, and elevated cholesterol or triglyceride levels [47]. The main factor influencing the development of metabolic syndrome is diet, which has been reported to affect the GM, including the GV [15].

Since the GM is a relevant player in the development of metabolic syndrome, it is reasonable to think that phages infecting these bacteria may also play an important role in metabolic syndrome by regulating such bacterial populations [49]. A recent study has shown that metabolic syndrome is associated with decreases in GV richness and diversity in a manner correlated with bacterial population patterns [49]. Dietary changes that cause a reduction in bacterial diversity have a direct consequence on GV diversity because there are bacterial species that are depleted from the GM and are therefore less accessible for predation by viruses. A recent study found that phages infecting Ruminococcaceae, Clostridiaceae, Bacteroidaceae, and Streptococcaceae predominated in the GV of patients with metabolic syndrome, whereas Bifidobacteriaceae phages were less abundant in patients with metabolic syndrome than in control samples [49]. Such results could reflect unequal predation by phages among the corresponding bacterial families in the gut [49]. This fact is interesting because bacteria of the genus *Bifidobacterium* inhibit the colonization of harmful intestinal bacteria, regulate the immune system, and exhibit anti-obesity and anti-inflammatory activities, thus preventing the progression of metabolic syndrome [57]. The identification of Bifidobacteriaceae species and their phages as more abundant among healthy controls is in line with established studies showing the depletion of these families in metabolic syndrome [58] and disease states associated with metabolic syndrome [59].

Furthermore, viral phages were significantly more prevalent in the GV of controls than in metabolic syndrome patients [49]. This apparent depletion of viral phages in GVs from metabolic syndrome patients may indicate a decrease in their infectivity and could be considered a link between this prominent human gut phage order and a disease state [49]. In contrast to what was reported by [49], the richness and diversity of the GV of children with metabolic syndrome were higher than those of normal-weight children without metabolic syndrome, along with an increased abundance of *Myoviridae* [47].

**Table 2 nutrients-15-00977-t002:** Research works investigating the relationship between the gut virome and metabolic diseases.

Disease/Model	Subjects	Determination	Main Findings	References
Obesity/humans	128 obese subjects and 101 lean subjects	Gut virome (GV), bacteriome, and viral–bacterial correlations	-Obese subjects, especially those with type 2 diabetes (T2D), had a lower gut viral richness and diversity than lean controls. -GV may play an important role in the development of obesity and T2D. -Eleven viruses, including *Escherichia* phage, *Geobacillus* phage, and *Lactobacillus* phage, were higher in obese subjects than in lean controls.	[28]
Inflammatory Bowel disease (IBD)/humans	12 household controls, 18 Crohn’s disease patients, and 42 ulcerative colitis patients	Stool samples investigated by virus-like particle enrichment and sequencing as well as bacterial 16S rRNA gene analysis	-Patients with IBD showed a significant increase in *Caudovirales* bacteriophages in their GV. -Changes in the GV may contribute to intestinal inflammation and bacterial dysbiosis.	[33]
Cirrhosis and hepatic encephalopathy/humans	40 controls and 163 cirrhotic patients	Stool metagenomics for bacteria and phages were analyzed in controls versus cirrhosis, within cirrhotic, hospitalized/not, and pre/post rifaximin	-Bacterial α-/β-diversity worsened from controls through cirrhosis patients. Phage α-diversity was similar in both groups. -No changes in α-/β-diversity of phages or bacteria were seen after postrifaximin treatment in cirrhotic patients.	[47]
Obesity and metabolic syndrome/humans	28 school-aged children (10 with normal weight, 10 obese, and 8 obese + metabolic syndrome)	Characterization of the gut DNA virome using metagenomic sequencing	-Phage richness and diversity of individuals with obese and obese + metabolic syndrome tended to increase with respect to controls. -The abundance of some phages correlated with gut bacterial taxa and with anthropometric and biochemical parameters altered in obese and obese + metabolic syndrome.	[50]
Bile acid metabolism/mice	7 germ-free C57BL/6J mice	Phage-induced repression of a tryptophan-rich sensory protein and repression of bile acid deconjugation	-Phages’ presence in the gut can affect the microbial metabolism of bile acids. -Phague BV01 and other phages from the family *Salyersviridae* are ubiquitous in the human gut, can infect a broad range of Bacteroides hosts, and affect bile acid metabolism.	[24]
Cerebral ischemia/mice	6 adult C57BL/6J mice	Determination of GV composition by shotgun metagenomics in fecal samples	-Following focal ischemia, the abundances of two viral taxa decreased, and those of five viral taxa increased compared with previous cohorts. -Abundances of Clostridia-like phages and Erysipelatoclostridiaceae-like phages were decreased in the stroke compared with previous cohorts	[36]
IBD/humans	40 fecal samples	Stool samples investigated by bioinformatics viral sequencing and bacterial 16S rRNA gene analysis	-Changes in GV and increased numbers of temperate phage sequences were found in individuals with Crohn’s disease. -Incorporating both bacteriome and GV composition offered better discrimination power between health and disease.	[49]
Metabolic syndrome/humans	196 participants with metabolic syndrome preceding cardiometabolic disease	Bulk whole genome and virus-like particle communities	-GV from metabolic syndrome patients exhibited low richness and diversity. -Viral clusters revealed that *Candidatus Heliusviridae*, a highly widespread gut phage, was found in >90% of metabolic syndrome patients.	[37]
Environmental enteric dysfunction and low growth rate/humans	94 children without diarrhea or human immunodeficiency virus	Gut bacterial and GV sequencing and analysis	- Three differentially abundant phages were identified in GV, depending on child growth velocities. -A positive correlation was found between bacteria and bacteriophage richness in children with subsequent adequate/moderate growth.	[35]
IBD/humans	Fecal samples from 24 children, 12 with inflammatory bowel disease and 12 controls	Identification of viral sequences and bacterial microbiota sequencing	-*Caudovirales*’ relative abundance was greater than that of *Microviridae* in both inflammatory bowel disease patients and healthy controls. -*Caudovirales* was more abundant in Chron´s disease patients than in ulcerative colitis patients, but not than in control patients. -Pediatric inflammatory bowel disease patients can be distinguished from healthy controls by bacterial community composition.	[34]
Crohn´s disease/mice	12–23 BALB/CYJ mice	Disruption of normal resident microbiota with streptomycin sulphate administration and phage therapy	-A single day of treatment with a phage cocktail significantly decreased the number of adherent invasive *Escherichia coli* in feces. -A single dose of the phage cocktail reduced dextran sodium sulphate-induced colitis symptoms in mice.	[60]
Colorectal cancer (CRC) and colonic adenoma/humans	71 colorectal cancer patients, 63 adenoma patients, and 91 healthy controls	Metagenomic sequencing of the gut microbiome and microbial interactions in adenoma and colorectal cancer patients	-Uncultured CrAssphage was higher in healthy controls and positively associated with beneficial butyrate-producing bacteria in gut microbiota (GM). -GV was much more dynamic than the GM as the disease progressed.	[52]
CRC/humans	90 human subjects, (30 healthy controls, 30 of whom had adenomas, and 30 of whom had carcinomas)	Stool samples analyzed by 16S rRNA gene, whole shotgun metagenomics, and purified virus metagenomic sequencing	-The CRC-associated GV consisted primarily of temperate bacteriophages. -Phages influenced cancer by directly modulating the influential bacteria.	[39]
Enteric pathogens/mice	100 C57BL/6J mice	Viruses generated from molecular clones were used to infect cell lines to liberate virions. Subsequently, clones were used to infect mice that were euthanized and investigated for results of viral infections	-Chronic murine astrovirus complements defects in adaptive immunity by elevating cell-intrinsic IFN-λ in the intestinal epithelial barrier in immunodeficient mice. -Elements of the GV can protect against enteric pathogens in an immunodeficient host.	[51]
Alcoholic hepatitis/humans	89 patients with alcoholic hepatitis, 36 with alcohol use disorder, and 17 healthy people as controls	Metagenomic sequencing of virus-like particles from fecal samples, fractionated using differential filtration techniques	-Patients with alcohol use disorder showed increased viral diversity in fecal samples compared to controls and patients with alcoholic hepatitis. -History of antibiotic treatment was associated with higher GV diversity. -Specific viral taxa, such as *Staphylococcus* phages and *Herpesviridae*, were associated with increased hepatic disease severity.	[1]
Viral entities/humans	662 samples from 1-year-old children	Processing of metagenomics and metaviromics datasets	-Viral enrichment during sample processing showed a loss of a significant part of the GV and did not represent integrated bacteria containing dormant phages (prophagues). -Approximately 65–83% of the viral populations in the metavirome were not aligned with the metagenome data.	[61]
Nonalcoholic fatty liver diseases (NAFLD)/humans	73 patients with NAFLD	RNA and DNA virus-like particles from fecal samples	-Patients with NAFLD and cirrhosis showed a significant decrease in intestinal viral diversity compared with controls. -Advanced NAFLD was associated with a reduction in the proportion of phages compared with other intestinal viruses.	[62]
IBD/humans	54 Patients with IBD and 23 healthy controls	Virus-like particles were purified from stool samples and characterized by DNA and RNA sequencing and VLP particle counts	-Viral populations associated with IBD showed perturbations with respect to healthy controls. -*Anelloviridae* showed a higher prevalence in IBD compared to healthy controls, and *Analloviridae* DNA levels were biomarkers of the effectiveness of immunosuppression. -IBD subjects had a higher ratio of *Caudovirales* to *Microviridae* phages compared to healthy controls.	[54]
CRC/humans	80 colorectal primary tumors tissues and corresponding normal colorectal tissues	GV and bacteriome analysis for CRC tissues	-The number of viral species increased whereas bacterial species decreased in CRC tissues compared with healthy ones. -Phages were the most preponderant viral species in CRC tissues, and the main families were *Myoviridae*, *Siphoviridae*, and *Podoviridae*. -Primary CRC tissues were enriched for Enterobacteria, *Bacillus*, *Proteus*, and *Streptococcus* phages, together with their pathogenic hosts in contrast to normal tissues.	[45]
Type 2 diabetes (T2D)/humans	71 T2D patients and 74 healthy controls	Whole-community metagenomic sequencing data of fecal samples	-Significant increase in the number of gut phages in fecal samples was found in the T2D group. -Significant alterations of the gut phageome cannot be explained simply by covariation with the altered bacterial hosts.	[46]
Cognitive maintenance/humans	120 subjects, 60 with obesity and 60 without obesity	Neuropsychological assessment in humans, extraction of fecal genomic DNA and whole-genome shotgun sequencing	-GV was dominated by *Caudovirales* and *Microviridae* phages. -Subjects with increased *Caudovirales* and *Siphoviridae* levels in the gut microbiome performed better cognitive status. -Phages should be considered novel actors in the microbiome–brain axis.	[46]
Cognitive maintenance/mice	11 mice were orally gavaged with saline and fecal material from humans	Behavioral testing in mice and study of gene expression in mouse prefrontal cortex	-Microbiota transplantation from human donors with increased specific *Caudovirales* levels led to increased scores in novel object recognition. -Phages should be considered novel actors in the microbiome–brain axis.	[53]
CRC/humans	74 patients with CRC and 92 healthy controls	Shotgun metagenomic analyses of viromes of fecal samples	-Gut phage community diversity was significantly increased in patients with CRC compared with controls. -GV dysbiosis was associated with early- and late-stage CRC.	[63]
Fructose intake/mice	25 C57BL/6J mice per group were used for phage production, and 36 mice were used for the in vivo dietary crossover study	*Lactobacillus reuteri* survival and phage production during gastrointestinal transit in mice	-Fructose intake activated the Ack pathway, involved in generating acetic acid, which promotes phage production.	[64]
Malnutrition/humans	8 monozygotic and 12 dizygotic twin pairs	Shotgun pyrosequencing of VLP-derived DNA	-Phage plus members of the *Anelloviridae* and *Circoviridae* families of eukaryotic viruses discriminate discordant from concordant healthy pairs.	[43]
Type 1 diabetes (T1D)/humans	103 T1D children and their mothers	Determination of virus antibodies, enterovirus RNA, and enzyme immunoassay analysis	-Autoantibody-positive children had more enterovirus infections than autoantibody-negative children before the appearance of autoantibodies. -Enterovirus infections seem to be associated with the induction of β-cell autoimmunity in young children with increased genetic susceptibility to T1D.	[65]
High-fat diet/mice	12 C57BL/6J pregnant female mice	Mice were administered with subtherapeutic antibiotic dosages or no antibiotic and subsequently analyzed for GV composition and 16S rRNA metagenomics	-High-fat diet significant shift away from the relatively abundant *Siphoviridae*, accompanied by increases in phages from the *Microviridae* family. -Phage structural genes significantly decreased after the transition to a high-fat diet.	[41]
IBD/humans and mice	Fecal samples collected from 3 ulcerative colitis patients in remission and 3 unrelated healthy controls were transferred to C57BL/6 mice	Fecal virus-like particles (VLPs) isolated from ulcerative colitis patients and healthy controls were transferred to mice	-VLPs isolated from ulcerative colitis patients specifically altered the relative abundances of several bacterial taxa involved in IBD progression in mice. -Phages are dynamic regulators of GM and implicate the GV in modulating intestinal inflammation and disease.	[31]
T1D/humans	Fecal samples from 11 children who had developed serum autoantibodies associated with T1D and healthy controls	Detection of phage and eukaryotic viral sequences	-GV of T1D subjects was less diverse than those of controls. Lower phage diversity in cases than in controls. -Specific components of the GV were both directly and inversely associated with the development of human autoimmune disease. -Among eukaryotic viruses, there was a significant enrichment of *Circoviridae*-related sequences in controls in comparison with T1D patients.	[44]
Hypertension/humans	196 samples	Viral and bacterial metagenomic investigation of fecal samples	-Virus could have higher discrimination power than bacteria to differentiate healthy prehypertension samples from hypertension patients	[48]

### 4.2. Obesity, Diabetes and Malnutrition

Obesity and diabetes are two forms of metabolic diseases that are highly prevalent worldwide [31]. In recent decades, there has been substantial evidence that abnormalities in GM composition can play a major role in the development of both diseases, although most evidence refers to gut bacterial composition and activities [66]. However, recent findings found significant differences in some viral families between obese and diabetic patients with respect to healthy patients in children [43,44] and mouse models [67].

A recent study found that both viral richness and diversity in the GV were lower than those found for lean subjects and in obese patients with T2D compared to lean controls [31]. Surprisingly, these results are contradictory to those previously reported by Ma et al. [45], who found a higher phage richness in T2D patients than in nondiabetic controls, as well as an increased relative abundance of the families *Siphoviridae*, *Podoviridae*, and *Myoviridae* and the unclassified order *Caudovirales* in T2D patients [45]. Previous Enterovirus infection was found to be a risk factor for T1D in children [43]. Afterward, another study showed a higher prevalence of the families *Circoviridae* and *Picornaviridae* in T1D pediatric patients than in healthy children [44]

High-fat-diet-induced obese mice showed a significant reduction in the family *Siphoviriade* and an increase in the virus families *Microviridae*, *Phycodnaviridae*, and *Miniviridae* in the fecal virome [65]. Rasmussen et al. [67] proposed GV modification as a potential therapeutic strategy against T1D and obesity. To verify this hypothesis, VLPs were transferred from slim mice to high-fat diet-induced obese mice, and as a result, weight gain and diabetes symptoms significantly decreased in obese mice [67].

Regarding viral species, 17 were found to have significantly different proportions in obese and diabetic subjects compared with lean subjects [65]. Among them, 4 viral species (*Micromonas pusilla* virus, *Cellulophaga* phage, *Bacteroides* phage, and *Halovirus,* unclassified DNA viruses) were higher in obese and T2D patients, whereas 13 viral species, including Hokovirus, Klosneuvirus, and Catovirus, were lower in obese-plus-T2D subjects with respect to lean controls [31,65].

Malnutrition is a global health problem that affects large numbers of individuals regardless of age, gender, race, social status, and geographic boundaries. It can be defined as an imbalance between energy and nutrient intake and the individual’s requirements, which can alter body measurements, compositions, and functions [68]. Children with malnutrition have been reported to have an immature gut GM composition compared to those without malnutrition. This lack of maturity in their GM is characterized by a lower α-diversity of the GM as well as a disproportionate expansion of the phylum Proteobacteria [69]. Similarly, disruption of the GV, including that of intestinal phages and eukaryotic virus members, could increase the risk of severe acute malnutrition [64]. A recent study found that phages of the order *Caudovirales* contributed differentially to stunted growth in malnutrition induced by environmental enteric dysfunction [37]. As the phylum Proteobacteria exists in a higher proportion in the GM with malnutrition relative to that of children without stunting and as *Caudovirales* phages (especially *Siphoviridae*) have Proteobacteria as one of the main bacterial hosts [70] and are also present in greater numbers in malnourished children than in healthy children, there might be a cooccurring phage-bacterial dynamic in the gut of stunted children [14], with both viruses/phages contributing to the severity of malnutrition.

### 4.3. Liver Diseases

The liver is a very important pivotal organ for host metabolism and maintains bidirectional communication with the gut via the gut–liver axis [61]. Thus, the liver plays a central role in the pathogenesis of several metabolic diseases. Recent works have investigated the potential changes in GV linked to liver diseases such as alcoholic hepatitis [51], NAFLD [61], and bile acid metabolism [50]. Additionally, although it is not a liver disease itself, the potential changes in the GV in response to the high intake of fructose are also important [63]. Beyond its lipogenic effect, fructose intake is also related to hepatic inflammation and cellular stress, such as oxidative and endoplasmic stress, which contributes to the progression of simple steatosis to nonalcoholic fatty liver disease [71].

In the case of NAFLD, patients with a more severe disease showed lower viral diversity than patients with a lower degree of disease or healthy controls [61]. At the same time, the proportion of phages among the total GV was also significantly lower in the case of severe NAFLD patients than in the less severe cases of the controls [61].

Regarding fructose intake, fructose increases the growth of *Lactobacillus reuteri*, a key important bacterial species considered an important lysogen, which are bacterial prophages inserted within their genomes that promote phage production [63]. Due to its higher sweetening power, fructose is one of the most abundant sugars consumed in a Western-style diet and results in more pronounced fructose-mediated phage production by *L*. *reuteri* than the intake of other sugars [63].

In the case of alcoholic liver disease, disease-specific alterations in the GV were reported, and gut viruses were identified as potent drivers of alcohol-specific liver disease [51]. In contrast to NAFLD, in alcoholic liver disease, increased viral diversity was found in patients with alcoholic liver disease, especially in those with a higher degree of alcoholic hepatitis [51]. Regarding viral proportions, the authors found an increase in eukaryotic viruses such as *Parvoviridae* and *Herpesviridae*, along with increases in intestinal phages such as Enterobacteriaceae phages, *Escherichia* phages, and *Enterococcus* phages in patients with alcoholic liver disease compared to controls [51]. Both *Parvoviridae* and *Herpesviridae* may be found in higher proportions in NAFLD subjects because they may have a depressed immune system or because the medication administered to them indirectly causes increased replication of the viruses in host cells [51]. The latter aspect regarding the relation of GV and hepatic disease is the relation of the activity of the *Bacteroides* phage BV01, a temperate phage integrated into *Bacteroides vulgatus,* a species that can repress the microbial modification of the bile acid pool in the host, which could be linked to beneficial changes in human host metabolism [50].

### 4.4. Cancer

Although the relationship between the GV and some types of cancer, such as metastatic melanoma [56] or adenoma [60], has been investigated, most works on the relationship between the human GV and cancer have focused on colorectal cancer (CRC) [52,53,54,55,60], which is logical since it is the type of cancer that has the most direct contact with the intestinal virome. According to Wong and Yu [72], CRC is related to modifications in the GM, in which some bacterial genera, such as *Roseburia*, are potentially protective taxa, whereas other genera, such as *Bacteroides, Escherichia, Fusobacterium*, and *Porphyromona,* are considered procarcinogenic agents.

Metagenomic analysis of stool samples from CRC patients revealed an increase in the richness and diversity of the intestinal GV with respect to control patients [53,54,60]. In another case, it was found that the differences between CRC patients and controls were insufficient for identifying specific virome communities between healthy and cancerous states [52]. The fact that phage richness is higher in CRC patients was hypothesized to be due to an increase in intestinal permeability, known as a “leaky gut”, caused by this phage, which facilitates the infiltration of pathogens and triggers chronic inflammation [73]. Another study found that phages, especially those from the families *Siphoviridae* and *Myoviridae*, are vital driving factors during the transformation from a healthy intestine to intestinal adenocarcinoma and to CRC [49].

In another work, the families Inovirus and Tunalikerirus were related to the development of CRC due to their capacity to insert random oligonucleotides into the bacterial genome, stimulating the production of bacterial biofilms and thus contributing to the carcinogenesis of the colon [74]. Both families are known to infect gram-negative bacterial hosts, including enterotoxigenic *Bacteroides fragilis*, *Fusobacterium nucleatum*, and genotoxic *Escherichia coli*, bacterial species often implicated in CRC development [53].

Another recent study of the GV in bulk from CRC patients reported significant reductions in Enterobacteria phages and CrAssphages compared to healthy controls [60]. Some viral species were reported to have the potential to act as discriminant markers of CRC; Orthobunyavirus, Tunalikevirus, Phikzlikevirus, Betabaculovirus, and Sp6likevirus were the viral genera with significantly higher abundances in CRC patients than in control patients [53].

Upon investigating primary tumor tissues of CRC, phages were found to be the most preponderant viral species, and the main families were *Myoviridae*, *Siphoviridae*, and *Podoviridae* [54,75]. The most frequently detected eukaryotic viruses include human endogenous Retrovirus K113, human Herpesviruses 7 and 6B, Megavirus chilensis, Cytomegalovirus, and Epstein-Barr virus [54]. A higher relative presence of human papillomavirus was also found in CRC versus non-CRC tissues [76]. Additionally, it was also shown that Epstein-Barr virus infection could contribute to CRC development by inducing mutagenesis in intestinal cells [54].

## 5. Intestinal Diseases Mediated by Bacteria

One of the earliest applications of phages in human medicine was their use as tools to fight pathogenic or antimicrobial-resistant bacteria. By infecting bacteria, phages can significantly alter the GM, primarily by integrating into bacterial genomes as prophages (lysogeny) or by killing bacteria (lysis). Among these, an increase in phage lytic action is associated with decreased bacterial diversity in IBD [50]. By modulating the intestinal bacteriome, intestinal phages show promising therapeutic potential in several diseases beyond bacterial infections [26] as well as therapeutic options in the treatment of drug-resistant infections in humans [77].

Among digestive tract diseases, IBD is a chronic inflammatory disease that is subdivided into two categories: Ulcerative colitis (UC) and Crohn’s disease (CD). Although the exact cause of IBD remains unknown, it was hypothesized that an alteration of the GM is closely related to its pathogenesis and significantly increases the risk of this disease [5]. Previous works found, for both UC and CD patients, an increase in *Caudovirales* content and a reduction in the abundance of *Microviridae*, as well as higher GV richness than those found in control patients [5]. In another recent study, a predominance of temperate virions (mostly Caudoviral taxa) was observed in the GV of IBD patients, suggesting an increased lysogenic conversion of phages [36].

In addition to phages, a recent study also demonstrated that eukaryotic virus populations that inhabit the human intestinal mucosa can be different in IBD patients with respect to healthy controls [42]. At the family level, UC patients showed a higher relative abundance of *Pneumoviridae*, whereas the *Anelloviridae* family was observed in higher proportions in healthy subjects. At the genus level, UC patients showed a higher presence of the Orthopneumovirus genus than healthy controls, whereas they showed lower amounts of the giant viruses Coccolithovirus and Minivirus, as well as the vertebrate-infecting virus Orthopoxvirus [42]. Additionally, other studies also found that the *Anelloviridae* family was more prevalent in IBD mucosal samples than in healthy controls [28,78].

## 6. Therapies including Transfer of Gut Viruses

The human gut microbiome strongly influences various metabolic processes, such as digestion, the immune system, and endocrine functions [49]. Therefore, GV modification has shown great potential as a disease therapy through fecal microbiota transplantation (FMT), fecal viral transplantation (FVT), and phage therapy (PhT). These therapies provide the first tantalizing evidence that manipulation of the phageome may be an effective therapeutic strategy [79]. There are precedents for the successful use of such therapies in the treatment of antibiotic-induced intestinal dysbiosis [80], IBD [7,78,79,81], obesity [67,82], T2D [67], or even certain types of cancer [55,56] (Table 3).

FMT is one of the most effective and accepted approaches to modulating the GM by restoring gut microbiome homeostasis through the reintroduction of beneficial microbes from a healthy donor [20]. The viral mechanism of action contributing to FMT therapies involves tripartite mutualistic interactions among bacteriophages or eukaryotic viruses, bacteria, and the host [20]. This approach has been successfully employed in clinical trials for the treatment of diseases such as IBD [85]. In fact, the efficacy of FMT in the treatment of *C. difficile* infections is approximately 90% and is currently the most promising application of FMT [20]. Additionally, FMT has also been successfully employed as a treatment for other symptoms and diseases, such as in the restoration of dysbiosis originating from severe antimicrobial treatments [80], showing better results than probiotic administration [20]. In addition, FMT was also useful in treating metabolic diseases such as T1D [86], T2D [84], obesity combined with T2D [67], necrotizing enterocolitis, small intestinal bacterial growth, and post-antibiotic microbiome dysbiosis [7,20]. These results, especially those in humans [84,86], suggest that FMT can change the metabolic repertoire of bacterial/mammalian host communities and/or regulate the profile of metabolic gene expression in the bacteriome.

However, the use of FMT presents significant risks to the health of the recipient subject due to the potential presence of pathogens, particularly obligate and opportunistic bacterial pathogens. Thus, an alternative option that avoids this risk is the use of FVT in which both eukaryotic and bacterial cells are removed, whereas the entire viral portion of a fecal sample is provided to another host [79]. In this regard, it was reported that some changes in viral populations could be related to the development of pediatric T1D [43] as well as pediatric and adult inflammatory bowel disease, including the reproducible expansion of *Caudovirales* and the reduction of *Microviridae* [28,35,36,40,42]. Among the methods using filtered fecal transplantation, FVT removes fecal bacteria and thus decreases the risk of bacterial infection associated with FMT, although the recipient maintains certain risks to the recipient due to the potential transfer of unwanted eukaryotic viruses (Figure 2). A seminal study by Ott et al. [79] demonstrated that administration of a sterile fecal filtrate achieved successful remission in patients with *C. difficile* infection. However, it should be considered that although the presence of eukaryotic viruses in the gut is essential for good maintenance of intestinal microbial homeostasis and host immunity [20], the human gut can harbor numerous genera of potentially pathogenic viruses, such as papillomaviruses, herpesviruses, hepatitis viruses, bocaviruses, enteroviruses, rotaviruses, and sapoviruses [87]. Therefore, FVT could be potentially dangerous, particularly for immunocompromised hosts [5]. Thus, especially for these patients, thorough GV monitoring of the donor should be performed to avoid the potential health risks derived from fecal transplantation [5].

Another option is PhT, a method that uses only phages to treat bacterial infections, which has demonstrated advantages in treating pathogenic and/or drug-resistant bacterial infections [5]. Currently, there is evidence that phages can stimulate and modulate the immune system of the host by various mechanisms. Indeed, phages can colonize the intestinal mucus layer, directly bind to mucin glycoproteins via their capsis, and provide the mammalian host with a defense mechanism against bacteria trying to breach the intestinal barrier [83]. Phages can also induce innate defenses from the host against bacterial colonization, stimulating the production of inflammatory cytokines and activating dendritic cells and innate lymphoid cells to produce interferons [20]. Some exploratory studies found a curative role of PhT in the treatment of extraintestinal diseases [5], such as improving diabetes outcomes in mice [64], reducing necrotizing enterocolitis in piglets [82], and reducing *Clostridium difficile* infection symptoms in humans, toward the ability of gut phages to restrict pathobiont growth and to improve the richness of the GM [79].

Both FVT and PhT have been applied to reshape the dysbiotic gut microbiome caused by antibiotic treatments [80], as well as in the treatment of various metabolic diseases, such as metabolic syndrome [82], mental disorders [88], and malignant tumors [56]. The majority of PhT studies are still exploratory and have been conducted in mice, and their results need to be confirmed in humans [5]. However, recent studies revealed that the relationships between phages and bacterial populations might influence growth stunting in children [14,37]. It was found that a combined therapy using phages targeting *Fusobacterium nucleatum* (a tumor-causing bacterium) and irinotecan (an antitumor drug) effectively inhibited the growth of *F. nucleatum* but stimulated the proliferation of the butyrate-producing bacterium *Clostridium butyricum* in mice [55]. In addition, other work showed that administering rats with a bacteriophage cocktail against *Staphylococcus*, *Streptococcus*, *Proteus*, *Pseudomonas*, *Escherichia coli*, *Klebsiella pneumoniae*, and *Salmonella* spp. achieved increased intestinal permeability, weight loss, and reduced pathogen activity [73]. Another recent study showed that phage supplementation with Lactococcal 936 bacteriophage increased memory in flies [46]. The authors attributed this effect to phages of the *Siphoviridae* family, which are positively associated with cognition, suggesting a possible association with poststroke cognitive dysfunction [46].

However, the results from PhT were not positive in all cases. In this sense, it was reported that the transplantation of VLPs from UC patient feces to human microbiota-associated mice aggravated colitis in mice and worsened the bacterial taxa associated with IBD pathogenesis [41,79]. In another work, the administration of a cocktail of Enterobacteriaceae phages belonging to the order *Caudovirales* exacerbated intestinal inflammation and did not induce the lysis of any endogenous microbes [38]. Thus, a better understanding of broad phage activity is necessary prior to proposing this strategy for more widespread use than is currently the case.

## 7. Conclusions

Although bacteria represent the most abundant population and account for the most DNA in the GM, other populations, such as viruses, are also abundant and can interact with both host and bacteria in various forms. Thus, one of the causes for which prebiotics and/or probiotics sometimes show lower than expected effects, both in intensity and duration, in the correction of intestinal dysbiosis could be the interaction with other microorganisms from the GM, such as viruses. Here, we present the latest scientific evidence that some phages and eukaryotic viruses can affect not only diseases impacting the digestive system but also metabolic diseases such as metabolic syndrome, obesity, T1D and T2D, and even cognitive status. The major barriers to a good understanding of GV composition and functions are the low VLP alignment capability with datasets and bioinformatic tools, as well as the ease of the degradation of RNA viruses from decay during metagenomic sequencing. In addition, DNA viruses do not possess any equivalent to the bacterial 16S rRNA gene, which would allow their rapid identification. However, it is important to avoid barriers that currently limit GV determination. Thus, in the near future, stool metagenomic investigations should include the identification of bacteria and viruses and their correlation networks. With this approach, it would be possible to achieve a better understanding of the action of intestinal viruses on metabolic diseases, as well as the use of fecal or viral transfer tools in the prevention and/or treatment of these pathologies.

## Figures and Tables

**Figure 1 nutrients-15-00977-f001:**
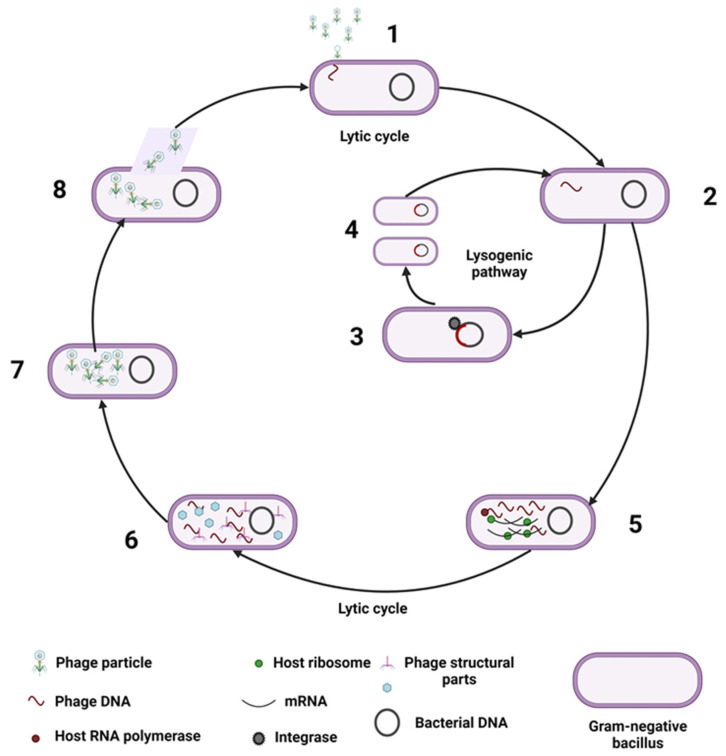
Representation of the main phage infection cycles. (1) Phage interaction with the bacterial surface through specific receptors and injecting its DNA into the host cell cytoplasm; (2) at this point, phages may follow the lysogenic cycle or the lytic cycle; (3) the phage genome integrates into the bacterial host genome using integrases; (4) the resulting bacteria containing dormant phages (prophage) replicate together with the bacterial genome for several generations and can enter the lytic cycle at any moment, e.g., under stress conditions; (5) following the lytic cycle, phages use bacterial molecular tools for protein synthesis and phage DNA polymerase for DNA replication; (6) the structural proteins are expressed; (7) new virions are formed; and (8) phage proteins, such as endoylsins and holins, break the cell membrane and allow the release of new viral particles ready for a new cycle of infections.

**Figure 2 nutrients-15-00977-f002:**
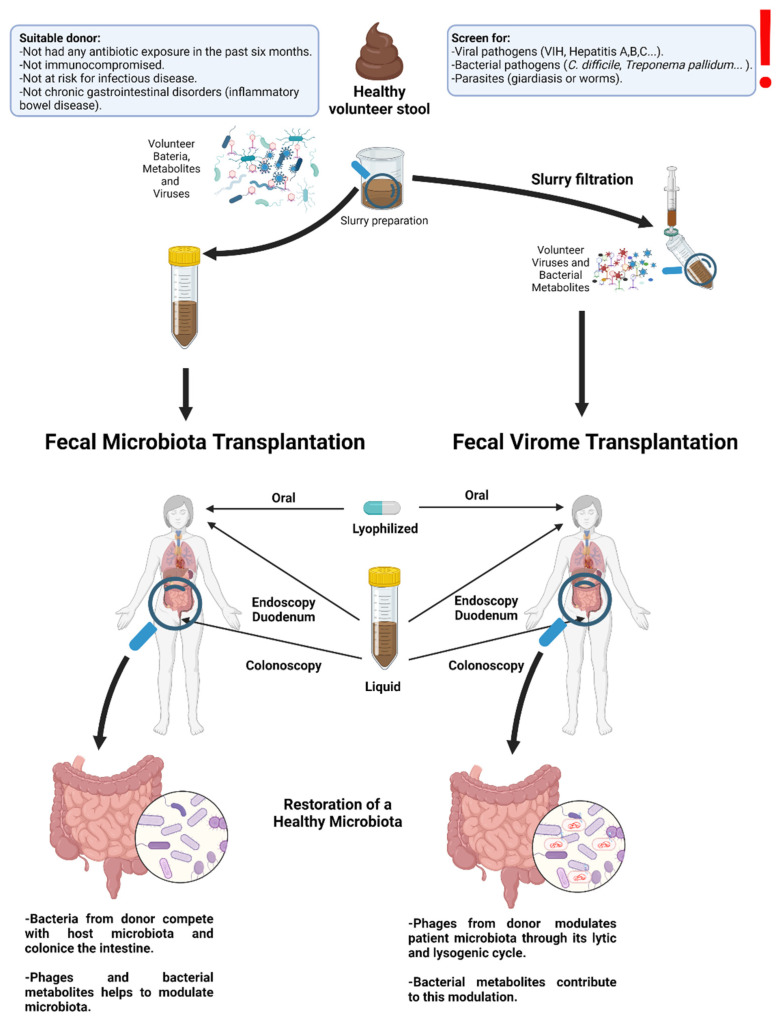
Representation of microbiota and virome transplantation in humans. The first step in this process is the selection of healthy volunteers and the creation of biobanks with stool samples. Volunteers must meet several requirements and have pathogen-free stool to avoid disease transmission to the transplant recipient. Two different strategies have been developed when performing fecal transplants. In fecal microbiota transplantation (FMT), no microorganisms are removed from the samples, whereas in fecal virome transplantation (FVT), the feces are filtered to eliminate the bacteria, molds, and yeasts present in the donor’s samples and to keep only the viruses, primarily bacteriophages, present in the fecal samples. Fecal transplants can be administered in a variety of ways, for example, through gastroscopy or colonoscopy. Stool samples can also be lyophilized, and therefore, these transplants can be performed orally, through the administration of a tablet, or through the anal route in the form of a suppository.

**Table 1 nutrients-15-00977-t001:** Research works on the metagenomic analysis of the human virome.

Model	Subjects	Determination	Main Findings	References
Humans (twin kids)	12 personal fecal samples (4 twins and his/her mothers)	Metagenomic sequencing of bacterial and viral content from fecal bulk	-The composition of the intestinal phageome has similarities in people from different parts of the world. -Ignorance of sequences in viral metagenomes can lead to errors in determining the diversity of the human virome. -This study identified and validated the genome sequence of CrAssphage.	[11]
Humans (healthy adults)	1986 individuals representing 16 countries	Metagenomic sequencing of the viral content of fecal bulk	-The variability in gut virome (GV) among studies due to technical deficiencies is greater than the effect of any disease. -Gut viral richness increases from birth to median age and declines in elderly individuals.	[13]
Human (1-year-old child)	662 paired samples obtained at 1 year from an unselected childhood cohort	Fecal sample metagenomics and comparison with metaviromics datasets	-An important part of the GV may be lost during the viral enrichment stage or may not be correctly detected as it is in the form of bacteria containing dormant phages (prophages). -Most viral populations in the metavirome were not found in the metagenome datasets.	[1]
Human (healthy children)	60 samples from 18 girls and 12 boys	Isolation and quantification of phages and bacteria from stool, metagenome assembly, and analysis	-Phages can regulate bacterial abundance and composition in an age-specific manner. -Phages could be related to the gut microbiota (GM) changes observed in child stunting.	[14]
Humans (twin kids)	8 infants (4 twin pairs)	Bulk virome characterization	-Both eukaryotic virome and bacterial microbiome expanded from birth to 2 years of age, whereas phageome composition decreased. -The infant microbiome is highly dynamic with respect to bacteria, viruses, and bacteriophages, whereas, in adults, it is more stable.	[7]
Humans (dietary intervention)	Purified virus-like particles from stool samples collected longitudinally from six healthy volunteers	Dietary intervention high-fat/low-fiber diet and comparison of the GV for 8 days	-Viral contigs were rich in functions required in lytic and lysogenic growth, as well as viral CRISPR arrays and genes for antibiotic resistance. -The largest source of variance among GV samples was interpersonal variation. -Dietary intervention caused a change in the GV community, causing convergence of GV in individuals with similar diets.	[15]
Mice (gnotobiotic)	5 Germfree C57BL/6 mice	Gnotobiotic mice subjected to predation by cognate lytic phages	-Shifts in the microbiome caused by phage predation alter the gut metabolome.	[16]
Humans (healthy adults)	10 healthy volunteers	Fecal samples were collected monthly and synchronously over a 12-month period	-Several groups of CrAss-like and *Microviridae* bacteriophages were identified as the most stable colonizers of the human gut. -There are stable, numerically predominant individual-specific persistent viromes typical of each subject.	[10]
Humans (healthy adults)	930 healthy adult subjects	Bulk DNA virome characterization	-Factors associated with urbanization and geography factors were the top covariates of GV variation. -GV showed more heterogeneity than the bacterial microbiome in the investigated samples.	[12]

**Table 3 nutrients-15-00977-t003:** Studies investigating fecal viral transplantation and its relationship with specific diseases.

Indication/Model	Subjects	Dosage and Time of Exposition	Main Findings	Reference
Inflammatory bowel disease (IBD)/humans	5 patients with symptomatic chronic-relapsing *Clostridium difficile* infection	Stool collection and characterization according to fecal microbiota transplantation (FMT) standards	-Fecal filtrate transfer (FFT) eliminated symptoms of *C. difficile* infection for a minimum period of 6 months. -Bacterial components, metabolites, or phages mediate many of the effects of FMT, and FFT might be an alternative approach, particularly for immunocompromised patients.	[79]
IBD/piglets	16 piglets were used to obtain FFT, transferred to 14 piglets by rectal transfer and to 13 by oro-gastric administration	FMT administration by cognate rectal FFT, oro-gastric FFT administration, and saline solutions	-FFT increased viral diversity and reduced Proteobacteria abundance in the ileal mucosa of FFT receiver piglets relative to controls.	[81]
Obesity and type 2 diabetes (T2D) induced by diet/mice	40 C57Bl/&NTac mice	Mice with a high-fat diet plus fecal viral transplantation (FVT) and high-fat diet plus ampicillin plus FVT were compared to controls	-At both 4 and 6 weeks after the first FVT, a significantly lower body weight gain was observed in the high-fat diet + FVT mice and compared to high-fat diet mice. -FVT normalized the blood glucose tolerance in the high-fat + FVT mice. -FVT strongly influences and partly reshapes the gut microbiota composition both with and without ampicillin treatment.	[67]
Phage adherence/in vitro	Bacteriophage T4	T4 phages were serially diluted and used to inoculate plates	-Phage adherence to the mucus model provides immunity applicable to mucosal surfaces. -The symbiotic relationship between phage and hosts provides protection for mucosal surfaces.	[83]
IBD/mice	C57BL/6J and C3H/HeJBir wild-type and Il10 mice kept under special pathogen-free conditions	Norovirus infection and investigation in changes induced in structural and functional intestinal barrier changes	-Norovirus caused epithelial barrier disruption in Il10 mice. -Norovirus might trigger individuals with a nonsymptomatic predisposition for IBD by impairment of the intestinal mucosa.	[78]
Antibiotic disturbance/mice	16 BALB/c mice	Administration of antibiotic treatment in the drinking water for 2 days	-Mice showed a perturbed microbiome because of antibiotic treatment, which was reverted over time similar to the pretreatment one. -Mice that had received FVT maintained gut microbiota (GM) more similar to the original before antibiotic treatment compared to mice that had received nonviable phages.	[80]
Obesity/humans	A total of 87 individuals took part-565 individuals responded to advertisements	Fecal microbiome transfer	-There was no effect of FMT on weight loss in adolescents with obesity, although a reduction in abdominal adiposity was observed.	[82]
*Clostridium difficile* infection/mice	26 C57BL/6 mice	Comparison of effects of the fecal VLP fraction against conventional FMT on the ileal microbiome	-VLP fraction played a potential role in modifying the gut microbiome during dysbiosis. -In both recipient groups, transplantation of the fecal VLP fraction alone produced the same outcome as that of the whole FMT.	[7]
Melanoma/humans	10 patients with anti-PD-1-refractory metastatic melanoma	Fecal transfer by FMT	-FMT showed favorable effects in immune cell infiltrates and gene expression profiles in both the gut lamina propia and tumor microenvironment. -There were two partial responses and one complete response in melanoma patients after FMT.	[56]
Colorectal cancer/humans	72 patients with colorectal cancer and 52 healthy subjects	Elimination of *F. nucleatum* by phages	-Oral administration of the phage-guided irinotecan-loaded nanoparticles in piglets led to negligible changes in hemocyte counts, immunoglobulin, and histamine levels, as well as liver and renal functions. -Phage-guided nanotechnology for the modulation of the GM might inspire new approaches for the treatment of colorectal cancer.	[55]
Stunting/humans	15 nonstunted and 15 stunted children	Isolation of gut phages, sterilization, and cross-infection of gut bacteria community belonging to other children	-Gut phages can regulate gut bacterial abundance and composition in an age-specific manner. -Proteobacteria from non-stunted children increased in the presence of phages from younger stunted children.	[14]
Leaky gut/rats	5 Wistar rats	Phage cocktail was given to rats for 10 days	-Increased intestinal permeability may be induced by phages that affect GM.	[73]
Obesity and T2D/humans	61 patients	Fecal transfer by FMT from healthy donors	-FMT achieved ≥20% of lean-associated microbiota in obese with T2D patients	[84]
*Clostridium difficile* infection/humans	24 subjects with *Clostridium difficile* infection and 20 healthy controls	Ultradeep metagenomic sequencing of virus-like particle preparations and bacterial 16S rRNA sequencing	-Subjects with *Clostridium difficile* infection showed a significantly higher abundance of *Caudovirales* and lower *Caudovirales* diversity, richness, and evenness compared with healthy controls. -FMT decreased the abundance of *Caudovirales* in *Clostridium difficile* infection. Symptoms of infections decreased when most *Caudovirales* came from the donor and not from the recipient.	[42]
Intestinal inflammation and colitis/humans and mice	58 C57Bl/6 and Swiss Webster germfree mice 20 Patients with active ulcerative colitis	Three independent experiments with a total of n = 23 for vehicle-treated animals and n = 21 for bacteriophage-treated animals	-Treating germ-free mice with bacteriophages led to immune cell expansion in the gut. -Increasing bacteriophage levels exacerbated colitis via toll-like receptors 9 and IFN-γ stimulation. -Phages from active ulcerative colitis patients induced more IFN-γ compared to healthy individuals.	[38]

## Data Availability

Not applicable.

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
