# Peer review of "The Human Gut Virome and Its Relationship with Nontransmissible Chronic Diseases"

_nutrients, 2023, doi:10.3390/nu15040977_

Round 1
Reviewer 1 Report
In this review Ezzatpour et al. describe the presence of viruses – bacterial as well as eukaryotic – in the gut of healthy and diseased humans and animals used as disease models. The gut microbiota is in the meantime well known to influence many systemic effects of the host. The contribution of the intestinal virome however is usual ignored. Thus, the manuscript summarized important and essential information. It is well written and the information is systematically presented in the text and in well-structured tables. The manuscript is therefore worth to be published. I have a few points which might be considered by the authors be for publication.
1. Line 99: the authors claim that phages are specific for a particular species. However, some phages can infect closely related bacterial species. Please clarify.
2. Line 104: please explain also pseudolysogenic and bacterial budding.
3. Lines 210 and 227 and following: controversies are described. A controversy is also described in311 and 317; once the richness in CRC is depredated and in the other context richness is higher. If the authors know or are able to explain the reasons, they should do it.
4. In general: it is a bit frustrating that in most examples the association of viral richness or the absence of it is correlated with disease (lines 129, lines 194, lines 201, lines 291. Usually no attempt is made to explain the mechanism how the phages or eukaryotic viruses exert this effect. Also how these differences are initiated. Probably, very little is still known on this issue. However, it would be interesting what ideas the authors have on these problems.
5. Line 345: “of” should most likely be “or”.
Author Response
With respect to the comment “In this review Ezzatpour et al. describe the presence of viruses – bacterial as well as eukaryotic – in the gut of healthy and diseased humans and animals used as disease models. The gut microbiota is in the meantime well known to influence many systemic effects of the host. The contribution of the intestinal virome however is usual ignored. Thus, the manuscript summarized important and essential information. It is well written and the information is systematically presented in the text and in well-structured tables. The manuscript is therefore worth to be published. I have a few points which might be considered by the authors be for publication“
The authors sincerely appreciate the constructive and kind words of the reviewer.
With respect to the comment about “Line 99: the authors claim that phages are specific for a particular species. However, some phages can infect closely related bacterial species. Please clarify”
Thank you very much for your comment, we agree with your observation. Consequently, the words “specific species” was changed to “small number of closely related bacterial species”.
With respect to the comment about “Line 104: please explain also pseudolysogenic and bacterial budding. “
Thank you very much for your comment. According to your suggestions the following paragraph was added to the new version of the manuscript: “Another of the phage cycles is the pseudolysogeny, also called the stationary phase of the phage in the host cell. In this phase there is neither multiplication of the phage genome as in the lytic cycle nor replication synchronized with the cell cycle of the host cell as in the lysogenic cycle. This process usually takes place when the host cell encounters unfavorable conditions such as starvation and ends when the phage enters an actual lysogenic cycle or enters a lytic cycle when bacterial growth conditions improve. This cycle seems to play an important role in phage survival as bacteria in the natural environment often exhibit very slow growth or starvation [19]. Another cycle of phages is bacterial budding. This cycle is really interesting as phages are released through the bacterial cell membrane without causing lysis of the bacterium by a budding-like process, producing a chronic release of phages [20].”
Accordingly, two new references were added to the references list:
- Łoś, M.; Węgrzyn, G. Pseudolysogeny. Adv Virus Res 2012, 82, 339-349. DOI 10.1016/B978-0-12-394621-8.00019-4.
- Mäntynen, S.; Laanto, E.; Oksanen, H.M.; Poranen, M.M.; Díaz-Muñoz, S.L. Black box of phage–bacterium interactions: Ex-ploring alternative phage infection strategies. Open Biol 2021, 11(9), 210188. DOI 10.1098/rsob.210188.
With respect to the comment about Lines 210 and 227 and following: controversies are described. A controversy is also described in311 and 317; once the richness in CRC is depredated and in the other context richness is higher. If the authors know or are able to explain the reasons, they should do it.
Thank you for your comments. In the first time, in which was cited “Consequently, it is logical to think that variations in phages that can depredate such bacterial genera can affect CRC pathogenesis”, it was a personal opinion for authors and refers to the predation of specific species that may influence the pathogenesis of CRC. In line 317, the increase in richness refers to results obtained by other authors. Therefore, we believe that there is no controversy between the two paragraphs. However, in case it is confusing, in the corrected version of the manuscript, the sentence "Consequently, it is logical to think that variations in phages that can depredate such bacterial genera can affect CRC pathogenesis." has been deleted.
With respect to the comment about In general: it is a bit frustrating that in most examples the association of viral richness or the absence of it is correlated with disease (lines 129, lines 194, lines 201, lines 291. Usually, no attempt is made to explain the mechanism how the phages or eukaryotic viruses exert this effect. Also how these differences are initiated. Probably, very little is still known on this issue. However, it would be interesting what ideas the authors have on these problems.
Thank you for your comment. According to your suggestions the following paragraphs was added to the new version of the manuscript: “Dietary changes that cause a reduction in bacterial diversity have a direct consequence on GV diversity because there are bacterial species that are depleted from the GM and are therefore less accessible for predation by viruses.” On the other hand, it was also added “Both Parvoviridae and Herpesviridae may be found in higher proportion in NAFLD subjects because they may have a depressed immune system or because the medication administered to them indirectly causes increased replication of the viruses in host cells [48].”
With respect to the comment about “Line 345: “of” should most likely be “or”.”
Thank you very much for your comment. We agree and consequently the word “of” was changed to “or”
Reviewer 2 Report
Well researched and well-written review focused on a very relevant topic, especially in understanding host-microbial interactions in relation to infectious and non-communicable diseases.
The review is focused on the human gut virome, a largely understudied aspect of the human microbiome and its association human health. The authors further review additional that shape the gut virome such as diet, genetics, drugs, and geography. The authors also highlight barriers to understanding the composition and functions of gut viruses due to the low alignment capability with datasets and bioinformatic tools, as well as the ease of degradation of RNA viruses from decay during metagenomic sequencing. They also suggest that stool metagenomic analysis should be included in the identification of bacteria and viruses as well as their correlation networks to better understand the role of intestinal viruses on metabolic diseases. All these aspects are critical in the development of prevention and treatment interventions against gut pathologies.
Picked up something minor on line 97, I think it should read “Relationship between gut virome and host” or “Host-Gut virome interactions/relationships”.
Author Response
With respect to the comment about “Well researched and well written review focusing on a very relevant to topic especially in understanding host microbial interactions in relationship to infectious and non-communicable diseases.”
The authors sincerely appreciate the constructive and kind words of the reviewer.
With respect to the comment about “Picked up a something minor on line 97, I think it should read “Relationship between gut virome and host” or “Host-Gut virome interactions/relationships”.
Than you very much for your comment. We are agree and consequently the phase “Relations gut virome-host” was changed to “Host-gut virome interactions/relationships”.
Reviewer 3 Report
Ezzatpour et al presented a narrative review that summarizes some pieces of evidence on correlations between gut virome and non-transmissible diseases. While the topic is interesting and timely, the below concerns I have should be addressed before the manuscript can be accepted for publication.
Methodology: How the literature was reviewed was not described. Even in a narrative review, the methods used to identify and screen relevant papers should be included.
Conceptual
- The manuscript was supposed to be concerning human diseases, but the tables also summarizing mouse studies. Were those all gnotobiotic models used to study human gut virome?
- In the end of the Introduction, the authors pointed out available studies “… have rarely deepened our understanding of its regulatory mechanisms” and “the objectives of this manuscript were therefore to provide and update the state-of-the-art knowledge of these relationships and the therapeutic potential of using viruses for prevention.” However, regulatory mechanisms of the viral components were rarely, if not never, discussed in this manuscript; therefore, the background knowledge of gut virome as therapeutics was not properly laid out. On the other hand, the therapeutic potential was only discussed in a short section.
- Since viruses are traditionally not considered “microorganisms”, the very first sentence in the Introduction may be revised into
“a complex biological system that mainly consists of bacteria, fungi, archaea, and viruses” or,
“a complex biological system that consists of bacteria, fungi, archaea, and viruses, among other components”
- The authors mentioned twice that “viruses do not possess any equivalent to the bacterial 16S rRNA gene, which would allow their rapid identification”. This is not true for RNA viruses, which contain the conserved gene RNA‐dependent RNA polymerase (RdRP). Personally, the trickiest part of virome studies is the lack of a virome database with comprehensive annotations (taxonomy, host, etc). This might partly be attributed to less effort spent previously, and challenges of recovering viral genomes from metagenomic datasets (without VLP enrichment).
- The authors may want to clarify their focus being (bacterio)phageome or both phageome and eukaryotic virome.
Writing
- Line 32: “metabolic human diseases” should be “human metabolic diseases”
- Line 40: “phage” should be defined here instead of the sentence at Line 42.
- The contents could be written more concisely.
E.g., “Subsequently, GV tends to resemble what it will be during the adult stage, where GV is usually stable over time, paralleling stability in the cellular microbiome [4]. In this regard, recent studies have found that the GV in a healthy adult is relatively stable, with > 90% of recognizable viral contigs persisting in everyone over 1 year [1,4]” could be re-written as,
“Subsequently, GV tends to resemble what it will be during the adult stage. Paralleling stability in the cellular microbiome [4], adult GV is usually stable over time, as evident from recent studies showing that > 90% of recognizable viral contigs persisted in individuals over one year [1,4]”
- Line 86: “Regarding all factors” could be removed.
- Subtitle “Relationships gut virome-host” is misleading. Besides, this section seems to include both “bacterial host” and “human host”, which is confusing.
- Line 149: “Viral assembly” could be interpreted as “the production of viruses” or “the assembly (development) of viral components in the gut microbiota”, but not be used to describe “the assembly of viral genomes/viral genome assembly”.
- Lines 211-217 are confusing. It seems that diversity was the focus here, which was not consistent in the two studies. But what other results were “compatible”?
- Line 234: “regarding viral families” was followed by “viral species”.
- Line 261 and Line 287: Why “Thus” was used to link the two sentences?
- Line 286: the definition of “prophage” should be moved to somewhere at the beginning.
- Line 385-386: The sentence “The viral mechanism of action contributing to FMT therapies involves tripartite mutualistic interactions among bacteriophages or eukaryotic viruses, bacteria, and the host [18]” disturbs the flow and may be relocated.
- Overall, the flow was not well maintained. This is most evident in the sub-section “4.2 Obesity, malnutrition and diabetes”, where paragraphs were created arbitually.
Author Response
With respect to the comments from the Reviewer 3:
With respect to the comment about “Ezzatpour et al presented a narrative review that summarizes some pieces of evidence on correlations between gut virome and non-transmissible diseases. While the topic is interesting and timely, the below concerns I have should be addressed before the manuscript can be accepted for publication.”
The authors sincerely appreciate the constructive and kind words of the reviewer
With respect to the comment about “Methodology: How the literature was reviewed With respect to the comment about as not described. Even in a narrative review, the methods used to identify and screen relevant papers should be included.”
Thank you very much for your comment, we agree with your observation. Consequently, At the end of the Introduction section, it was included the following paragraph: “To achieve this goal, A narrative literature search was conducted up to 10 July 2021 for databases Web of Science and Scopus. The term “human gut virome” was searched in the field “title, abstract and keywords” in the case of Scopus and “topic” in the case of Web of Science. A total of 269 articles were found. The selection of articles will be limited to studies published in English, with no restrictions on the year of publication, although the most prominent articles are those published after 2018. The authors reviewed the titles and the abstracts. If the abstracts reported useful information, full texts were read, and if the preestablished eligibility criteria were met, they were included in the review. After selecting the articles which falls into de selected scope, a total of 87 articles were selected and included in the review.”
With respect to the comment about “The manuscript was supposed to be concerning human diseases, but the tables also summarizing mouse studies. Were those all gnotobiotic models used to study human gut virome?”
Thank you for your comment. Not in all cases are animals used with gnotobiotic models, nor humanized, but in some of the assays animals with their own original characteristics are included. Although they do not have the same applicability as assays performed on humans, these assays are also useful for studying the effects of external factors on metabolic pathways or physiological factors that affect the development of metabolic diseases. We believe that the degree of confusion is limited since when gnotobiotic or ferm-free animals have been used, it has been specified so that the reader can distinguish them correctly.
With respect to the comment about “In the end of the Introduction, the authors pointed out available studies “… have rarely deepened our understanding of its regulatory mechanisms” and “the objectives of this manuscript were therefore to provide and update the state-of-the-art knowledge of these relationships and the therapeutic potential of using viruses for prevention.” However, regulatory mechanisms of the viral components were rarely, if not never, discussed in this manuscript; therefore, the background knowledge of gut virome as therapeutics was not properly laid out. On the other hand, the therapeutic potential was only discussed in a short section.”
Thank you for your comment. We consider that the Reviewer is right and consequently, the phrases “To date, most of the studies conducted in this regard have demonstrated only the association of GV and diseases but have rarely deepened our understanding of its regulatory mechanisms” as well as “these relationships and the therapeutic potential of using viruses for prevention” was deleted from the revised version of the manuscript.
With respect to the comment about “Since viruses are traditionally not considered “microorganisms”, the very first sentence in the Introduction may be revised into “a complex biological system that mainly consists of bacteria, fungi, archaea, and viruses” or, “a complex biological system that consists of bacteria, fungi, archaea, and viruses, among other components.”
Thank you for your comment. The phrase was changed to the second option suggested in the revised version of the manuscript.
With respect to the comment about The authors mentioned twice that “viruses do not possess any equivalent to the bacterial 16S rRNA gene, which would allow their rapid identification”. This is not true for RNA viruses, which contain the conserved gene RNA‐dependent RNA polymerase (RdRP). Personally, the trickiest part of virome studies is the lack of a virome database with comprehensive annotations (taxonomy, host, etc). This might partly be attributed to less effort spent previously, and challenges of recovering viral genomes from metagenomic datasets (without VLP enrichment).
Thank you for your comment. If fact, it was specified that the sentence refers only to DNA viruses. Additionally, the following paragraph was added: “RNA viruses contain the conserved gene RNA‐dependent RNA polymerase (RdRP), which allows a broad viral identification [25]. However, the lack of a virome database with comprehensive annotations is a major concern to achieve a broad identification of RNA virome.”
Consequently, the following reference was added to the references list:
Ferrero, D.; Ferrer-Orta, C.; Verdaguer, N. Viral RNA-dependent RNA polymerases: A structural overview. Subdell Biochem 2018, 88, 39-71. DOI 10.1007/978-981-10-8456-0_3.
With respect to the comment about “The authors may want to clarify their focus being (bacterio)phageome or both phageome and eukaryotic virome.”
Thank you for your comment. The term “phageome” was deleted in any cases that do not refer to a previous work that were cited in the text. Obviously, this manuscript is focused to both phages and eukaryotic viruses, although most works published, and consequently included in this manuscript are focused on phages.
With respect to the comment about “Line 32: “metabolic human diseases” should be “human metabolic diseases”
Thank you very much for your comment, we agree with your observation. Consequently, “metabolic human diseases was changed to “human metabolic diseases”.
With respect to the comment about “Line 40: “phage” should be defined here instead of the sentence at Line 42.”
Thank you very much for your comment, we agree with your observation. Consequently, the phrase “viruses that infect bacteria” was moved from line 42 to line 40.
With respect to the comment about “Subsequently, GV tends to resemble what it will be during the adult stage, where GV is usually stable over time, paralleling stability in the cellular microbiome [4]. In this regard, recent studies have found that the GV in a healthy adult is relatively stable, with > 90% of recognizable viral contigs persisting in everyone over 1 year [1,4]” could be re-written as, “Subsequently, GV tends to resemble what it will be during the adult stage. Paralleling stability in the cellular microbiome [4], adult GV is usually stable over time, as evident from recent studies showing that > 90% of recognizable viral contigs persisted in individuals over one year [1,4]”
Thank you very much for your comment, we agree with your observation. Consequently, the cited paragraph was changed to “Subsequently, GV tends to resemble what it will be during the adult stage, where GV is usually stable over time, paralleling stability in the cellular microbiome [4]. In this re-gard, recent studies have found that the GV in a healthy adult is relatively stable, with > 90% of recognizable viral contigs persisting in everyone over 1 year [1,4].”
With respect to the comment about “Line 86: “Regarding all factors” could be removed.”
Thank you very much for your comment, we agree with your observation. Consequently, the phrase “regarding all factors” was deleted from line 86.
With respect to the comment about “Subtitle “Relationships gut virome-host” is misleading. Besides, this section seems to include both “bacterial host” and “human host”, which is confusing.”
Thank you very much for your comment, we agree with your observation. The subtitle “Relationships gut virome-host” was changed to “Host-Gut virome interactions/relationships”. This change was made attending the suggestion from other Reviewer and we hope it will be easier to understand than the original.
With respect to the comment about “Line 149: “Viral assembly” could be interpreted as “the production of viruses” or “the assembly (development) of viral components in the gut microbiota”, but not be used to describe “the assembly of viral genomes/viral genome assembly”.
Thank you very much for your comment, we agree with your observation. The phrase “Viral assembly” was changed to “The assembly of viral genomes”.
With respect to the comment about Lines 211-217 are confusing. It seems that diversity was the focus here, which was not consistent in the two studies. But what other results were “compatible”?
Thank you very much for your comment, in fact the results obtained in the cited works carried out in children (Bikel et al., 2021) and adults (de Jonge et al., 2022) were not enterely “compatible”, because in one case (Bikel et al., 2021) phage richness and diversity was higher in obese and obese plus metabolic syndrome patients than in control patients, whereas in adults (de Jonge et al., 2022), phage richness was lower in obese patients than in controls. This fact was corrected in the revised version of the manuscript.
Subsequently, the paragraph “Overall, the results reported by [46] in adult patients are compatible with those previously reported in patients with metabolic syndrome among school-aged children [44], who found a lower abundance of CrAss-like phages in children with metabolic syndrome. However” was deleted from the revised version of the manuscript.
With respect to the comment about Line 234: “regarding viral families” was followed by “viral species”.
Thank you very much for your comment, we agree with your observation. Consequently, “Regarding viral families, 17 viral species” was changed to “Regarding viral species, 17…”.
With respect to the comment about Line 261 and Line 287: Why “Thus” was used to link the two sentences?
Thank you very much for your comment, we agree with your observation. Consequently, both “Thus” were deleted from the corrected version of the manuscript.
With respect to the comment about Line 286: the definition of “prophage” should be moved to somewhere at the beginning.
Thank you very much for your comment, we agree with your observation. Consequently, the definition of “prophage” was moved to the first time that appears in the main text. Additionally, it was added to Table 1, Table 2, and Figure 1 Caption.
With respect to the comment about Line 385-386: The sentence “The viral mechanism of action contributing to FMT therapies involves tripartite mutualistic interactions among bacteriophages or eukaryotic viruses, bacteria, and the host [18]” disturbs the flow and may be relocated.
Thank you very much for your comment, we agree with your observation. However, we think that the cited phrase would alters the flow in any place that will stated, and consequently, it was deleted from the revised version of the manuscript.
With respect to the comment about Line 385-386: The sentence Overall, the flow was not well maintained. This is most evident in the sub-section “4.2 Obesity, malnutrition and diabetes”, where paragraphs were created arbitually.
Thank you very much for your comment. In the version of the manuscript, the flow was changed to first describe obesity and diabetes, describing viral families in different proportions with respect to control patients, describing viral species involved. Afterwards it was described malnutrition and their relation with GV.
Round 2
Reviewer 3 Report
Thank the authors for addressing my concerns. I do think a few rounds of language checking are needed to finalize the manuscript. For example, there are two "a" in the very first sentence of the Introduction.
Author Response
Thank your vey much for your comment. Despite the original version was previously corrected by an professional Englis Editor, during the review process we could we may have made some mistakes. Addording to your suggestion, the word "a" was deleted from the first sentece of the introduction. Additionally, the whole manuscript was checked by the American Journal Experts digital editting tool. Thus, the manuscript was minor changed. The last cheeck in the cited platform releals an grammar rate of 8.8/10, being better than 95% of the manuscripts submitted to the platform. Attached is the result of the grammatical evaluation obtained in American Journal Experts
